# Evaluation of the Antiviral Efficacy of Subcutaneous Nafamostat Formulated with Glycyrrhizic Acid against SARS-CoV-2 in a Murine Model

**DOI:** 10.3390/ijms24119579

**Published:** 2023-05-31

**Authors:** Ju Hwan Jeong, Woong Hee Lee, Seong Cheol Min, Beom Kyu Kim, On Bi Park, Santosh Chokkakula, Seong Ju Ahn, Sol Oh, Ji-Hyun Park, Ji Won Jung, Ji Min Jung, Eung-Gook Kim, Min-Suk Song

**Affiliations:** 1Department of Microbiology, Chungbuk National University College of Medicine and Medical Research Institute, Cheongju-si 28644, Republic of Korea; 2MODNBIO Inc., Seoul 08378, Republic of Korea; 3Institute of Biotechnology, Chungnam National University, Daejeon 34134, Republic of Korea; 4Biomedical Engineering from the Department of Biotechnology, Graduate School of Advanced Fusion Technology, Cheongju University, Cheongju-si 28160, Republic of Korea; 5Department of Medical Engineering, Catholic University of Daegu (DCU), Gyeongsan-si 38430, Republic of Korea; 6Department of Medical IT, Catholic Kwandong University, Gangneung-si 25601, Republic of Korea; 7Department of Biochemistry, Chungbuk National University College of Medicine and Medical Research Institute, Cheongju 28644, Republic of Korea

**Keywords:** SARS-CoV-2, antiviral, nafamostat, glycyrrhizic acid, in vivo mouse model

## Abstract

The ongoing COVID-19 pandemic highlights the urgent need for effective antiviral agents and vaccines. Drug repositioning, which involves modifying existing drugs, offers a promising approach for expediting the development of novel therapeutics. In this study, we developed a new drug, MDB-MDB-601a-NM, by modifying the existing drug nafamostat (NM) with the incorporation of glycyrrhizic acid (GA). We assessed the pharmacokinetic profiles of MDB-601a-NM and nafamostat in Sprague-Dawley rats, revealing rapid clearance of nafamostat and sustained drug concentration of MDB-601a-NM after subcutaneous administration. Single-dose toxicity studies showed potential toxicity and persistent swelling at the injection site with high-dose administration of MDB-601a-NM. Furthermore, we evaluated the efficacy of MDB-601a-NM in protecting against SARS-CoV-2 infection using the K18 hACE-2 transgenic mouse model. Mice treated with 60 mg/kg and 100 mg/kg of MDB-601a-NM exhibited improved protectivity in terms of weight loss and survival rates compared to the nafamostat-treated group. Histopathological analysis revealed dose-dependent improvements in histopathological changes and enhanced inhibitory efficacy in MDB-601a-NM-treated groups. Notably, no viral replication was detected in the brain tissue when mice were treated with 60 mg/kg and 100 mg/kg of MDB-601a-NM. Our developed MDB-601a-NM, a modified Nafamostat with glycyrrhizic acid, shows improved protectivity against SARS-CoV-2 infection. Its sustained drug concentration after subcutaneous administration and dose-dependent improvements makes it a promising therapeutic option.

## 1. Introduction

In 2019, the SARS-CoV-2 virus emerged in Wuhan, China, and escalated to a pandemic in March 2020 [1]. As of 2023, it has resulted in approximately 600 million infections and 6 million fatalities worldwide, according to the WHO. The development and distribution of vaccines against the virus were significant breakthroughs; however, the emergence of new variants with escape mutations, such as the E484K mutation, has reduced the efficacy of these developed vaccines [2]. Consequently, there is a pressing need to develop and clinically evaluate novel antiviral drugs and repurpose clinically evaluated drugs to treat current and emerging SARS-CoV-2 variants, particularly following the prevalence of the Delta and Omicron variants. While monoclonal antibodies have been effective against the S protein of SARS-CoV-2, their antiviral effect may be insufficient in novel variants such as Omicron [3]. Direct-acting antiviral molecules, including remdesivir, nirmatrelvir, and molnupiravir, have been developed and approved for SARS-CoV-2 treatment. Remdesivir, a nucleoside analog initially developed for ebolavirus [4], was approved by the Food and Drug Administration (FDA) for SARS-CoV-2 therapy in 2020. Nirmatrelvir (Paxlovid^®^) and molnupiravir were granted emergency use authorization (EUA) in 2021, offering the advantage of oral administration over intravenous administration, a disadvantage of remdesivir. However, nirmatrelvir faces challenges as several mutations in Nsp 5 of SARS-CoV-2 have been reported for resistance to nirmatrelvir. Thus, host-directed functional antivirals have been suggested for treating SARS-CoV-2 that escape resistance mutations in direct-functional small molecules.

The SARS-CoV-2 virus, similar to SARS-CoV, binds to the angiotensin-converting enzyme 2 (ACE-2) receptor on the host membrane, initiating the entry process. During this entry, cell surface proteases such as TMPRSS2 cleave the Spike protein into the S1/S2 domain. Consequently, TMPRSS2, the host membrane surface protease, serves as a drug target for both SARS-CoV and SARS-CoV-2. Nafamostat mesylate, a synthetic serine protease inhibitor targeting TMPRSS2, was initially developed as a drug for pancreatitis. It has been reported to have potential therapeutic effects for antiviral and anti-cancer functions [5]. Recent studies identified nafamostat as an antiviral agent for SARS-CoV-2 [6], and it is currently undergoing clinical trials in several countries, including Korea, Japan, and Ukraine. However, nafamostat administration has limitations due to its poor pharmacokinetics, with a duration of only 8–23 min [7,8]. Researchers have developed and tested various antivirals for their inhibitory effects on SARS-CoV-2. Among them, natural compounds such as baicalin, curcumin, and hesperidin have been found to exhibit inhibitory effects on SARS-CoV-2 [9]. Another natural compound, glycyrrhizic acid (GA), extracted from licorice root, has demonstrated inhibitory effects on SARS-CoV by inhibiting the attachment of N-acetyl glucosamine to the S-protein’s carbohydrates [10]. GA has also been reported to have anti-inflammatory effects, regulating the PI3K-AKT, IL-17, and MAPK signaling pathways and inhibiting the TLR-4/NF-κB signaling pathway. These effects could be beneficial in treating SARS-CoV-2 infection [11]. In this study, we identified glycyrrhizic acid as a potential candidate for antiviral formulation agents. We previously developed and tested nafamostat using a drug delivery system (DDS) using micelle NPs against SARS-CoV-2 [12,13]. Furthermore, it was confirmed that a triblock structure (with a long hydrophilic chain, a hydrophobic segment, and a short hydrophilic chain) could form micelles and be used to encapsulate a hydrophilic drug [14]. We developed a novel formulation glycyrrhizic acid-formulated nafamostat (MDB-601a-NM), which improved the administration route and antiviral activity duration in a SARS-CoV-2 mouse model.

## 2. Results

### 2.1. Nafamostat Formulation with Glycyrrhizic Acid: Characterization of NM-Loaded Micelle NPs

Micelles formed with DB13751 were anticipated to exhibit stable drug delivery system (DDS) effects while consistently encapsulating Nafamostat (NM). Initially, DB13751 was dissolved in ethanol, followed by oil addition to adjust the critical micelle concentration, leading to micelle formation. Nafamostat was then loaded into the micelle via sonication, and the desired NM-loaded micelle nanoparticles (NPs) were obtained through lyophilization. Transmission electron microscopy (TEM) observation revealed that the NM-loaded micelle NPs displayed a spherical morphology (Figure 1a). Dynamic light scattering was employed to determine the particle sizes and distributions of NM-loaded micelle NPs, with the particle size of NM-loaded micelle NPs measuring approximately 220 nm (Figure 1b).

### 2.2. Pharmacokinetics and Single-Dose Toxicity Test of MDB-601a-NM in Sprague-Dawley Rats

Pharmacokinetics of MDB-601a-NM were evaluated in 6–8-week-old Sprague-Dawley rats through intravenous (IV) and subcutaneous (SC) administration of Nafamostat and MDB-601a-NM, with blood samples collected for drug concentration analysis (Figure 2a). Nafamostat exhibited rapid clearance in rats administered 1 mg/kg IV, with undetectable levels within 0.5 h. In rats administered 20 mg/kg SC, nafamostat was detected after 2 h. In contrast, MDB-601a-NM demonstrated sustained drug concentration in the sera, with increasing concentrations observed up to 0.033 h post-administration and persisting for up to 8 h at a dose of 50 mg/kg. Furthermore, MDB-601a-NM exhibited a longer in vivo half-life compared to nafamostat when administered subcutaneously, with drug concentration persisting for more than 2 h.

### 2.3. Evaluation of Single-Dose Toxicity of MDB-601a-NM in Sprague-Dawley Rats

To assess the single-dose toxicity of MDB-601a-NM in Sprague-Dawley rats, high concentrations (800, 2000, and 5000 mg/kg) of MDB-601a-NM were administered subcutaneously and subsequent changes in the body weights of the rats were measured over 15 days post-injection (Figure 2b). The body weight of naive rats demonstrated a steady increase over time, and rats administered with 800 and 2000 mg/kg concentrations of MDB-601a-NM exhibited a similar trend. However, rats administered with 5000 mg/kg concentration of MDB-601a-NM did not show an increase in body weight compared to their initial weight, indicating toxicity associated with high-dose administration.

Additionally, subcutaneous administration of high concentrations (800, 2000, and 5000 mg/kg) of MDB-601a-NM led to observed swelling at the injection site (Table 1). Rats administered with 800 mg/kg concentration of MDB-601a-NM exhibited swelling starting from 5 days post-injection, with signs of recovery at 11 days post-injection. In contrast, rats administered with 2000 and 5000 mg/kg concentrations of MDB-601a-NM displayed continuous swelling from 4 to 15 days post-injection. Male rats administered with 800 mg/kg concentration of MDB-601a-NM showed improved swelling 1 day earlier than female rats; however, no significant difference in swelling improvement was observed between male and female rats administered with 2000 and 5000 mg/kg concentrations of MDB-601a-NM.

### 2.4. In Vivo Protective Efficacy of MDB-601a-NM against SARS-CoV-2 Infection

To assess the protective efficacy of MDB-601a-NM against SARS-CoV-2, all treatment groups, including MDB-601a-NM, glycyrrhizic acid, and nafamostat, displayed weight loss after 4 days post-infection (DPI) and mortality in mice after 7 DPI (Figure 3a). Notably, the group administered 285 mg/kg of glycyrrhizic acid exhibited mortality in mice at 7 DPI, with all individuals succumbing to the infection by 9 DPI. As a positive control, the nafamostat administration group demonstrated continuous weight loss until 7 DPI, followed by a gradual recovery starting from 8 DPI, with 33% of the mice surviving (Figure 3b).

In the 20 mg/kg MDB-601a-NM administration group, 15% weight loss was observed, which did not recover by 8 DPI, and all individuals died. However, in the 60 mg/kg MDB-601a-NM administration group, 18% weight loss occurred until 9 DPI, with 50% of the mice surviving. In contrast, the 100 mg/kg MDB-601a-NM administration group showed a 15% weight loss until 8 DPI, followed by a gradual recovery until 13 DPI. Notably, at 11 DPI, 67% of the mice in the 100 mg/kg MDB-601a-NM group survived. Importantly, the vehicle group did not exhibit any weight loss or mortality due to SARS-CoV-2 infection.

### 2.5. Assessment of MDB-601a-NM Therapy in Inhibiting Viral Replication in SARS-CoV-2 Infected Tissue

To evaluate the protective effect of MDB-601a-NM, we measured viral replication in collected lung and brain tissues at 5 DPI using the TCID50 assay (Figure 3c). In infected lung tissues, all treatment groups (MDB-601a-NM, glycyrrhizic acid, and nafamostat) demonstrated high viral replication, except for the vehicle group. Notably, none of the MDB-601a-NM treated groups exhibited a significant decrease in viral replication compared with nafamostat-treated mice. In contrast, the vehicle group and the 60 mg/kg and 100 mg/kg MDB-601a-NM treated groups did not show detectable viral titers in brain tissues. Nafamostat, as the positive control group, reduced viral replication by 1.21 Log10TCID50/mL, but the difference was not statistically significant. Interestingly, the 20 mg/kg MDB-601a-NM treated group demonstrated a slight decrease in viral replication to 1.13 Log10TCID50/mL. Additionally, the 285 mg/kg glycyrrhizic acid-treated group exhibited lower viral replication compared with the mock-treated group.

### 2.6. Comparison of Histopathological Features in Mice Treated with MDB-601a-NM Therapy and Infected with SARS-CoV-2

We performed H&E and IHC staining of lung and brain tissues from mice treated with MDB-601a-NM at 5 DPI. The collected tissues were inactivated using a 10% formalin solution for 14 days. After staining, we evaluated preclinical symptoms such as severe lesions, destroyed cells, detected immunocytes, and the presence of SARS-CoV-2 antigen, which resulted from pneumonia (Figure 4). The lung and brain tissues of the vehicle group displayed clear and healthy lesions in both H&E and IHC staining. In contrast, the mock and nafamostat-treated groups exhibited severe lesions, with immunocytes present in alveoli and respiratory veins in H&E staining, and a large number of SARS-CoV-2 antigens detected in IHC staining. However, the MDB-601a-NM treated groups demonstrated histopathological differences that depended on the treatment dose. The 20 mg/kg group exhibited severe lesions similar to the mock-treated group and detected a large number of antigens in the lung and brain tissues. The 60 mg/kg group displayed moderate lesions and antigens, while the 100 mg/kg group showed clear alveoli with fewer immunocytes and antigens detected in lung and brain tissues. These results indicate that the protective efficacy of MDB-601a-NM is dependent on the treatment dose, and higher doses demonstrate improved efficacy.

## 3. Discussion

The ongoing COVID-19 pandemic continues to pose a significant global public health threat, necessitating the relentless development of antiviral agents and vaccines with the highest efficacy and requiring rigorous verification. However, through drug repositioning, the timeline for developing novel therapeutics can be significantly reduced, thus contributing to the protection of public health against the threat of COVID-19. In this context, we developed a new drug using a modified formulation of nafamostat, an existing drug, combined with glycyrrhizic acid. Subsequently, we validated the novel compound and its pharmacokinetic profile. This approach not only conserves time and resources needed for drug development but also increases the probability of success by building on the existing data of the parent compound. Therefore, drug repositioning emerges as a crucial strategy to expedite the development of innovative therapeutics for COVID-19 and other infectious diseases, offering a potential solution to the ongoing public health crisis.

The evaluated pharmacokinetic profiles provide valuable insights into the behavior of MDB-601a-NM and nafamostat in vivo. The rapid clearance of nafamostat upon intravenous administration raises concerns regarding its suitability for prolonged therapeutic use. Conversely, the sustained drug concentration of MDB-601a-NM after subcutaneous administration suggests its potential as a therapeutic agent with long-lasting effects. Results from single-dose toxicity studies indicate that high-dose administration of MDB-601a-NM via the subcutaneous route can cause toxicity, as evidenced by the lack of body weight increase in rats administered with a 5000 mg/kg concentration of MDB-601a-NM. Additionally, the injection site of rats administered with high concentrations of MDB-601a-NM exhibited persistent swelling, highlighting the need for careful observation and monitoring. Further studies on the pharmacokinetics of MDB-601a-NM, such as the correlation between drug concentration and therapeutic efficacy, are essential to fully comprehend its therapeutic potential.

We evaluated the efficacy of MDB-601a-NM and nafamostat in inhibiting viral replication and protecting against SARS-CoV-2 infection in the K18 hACE-2 transgenic mouse model. Although 20 mg/kg of MDB-601a-NM did not protect against the 5MLD50 of SARS-CoV-2, 60 mg/kg and 100 mg/kg of MDB-601a-NM demonstrated improved protectivity in terms of weight loss and survival rates compared to the lower dose. In contrast, the survival rate of nafamostat-treated mice was only 33% in SARS-CoV-2-infected mice, while 60 mg/kg and 100 mg/kg of MDB-601a-NM yielded survival rates of 50% and 67%, respectively. Moreover, the 60 mg/kg and 100 mg/kg doses of MDB-601a-NM significantly decreased viral replication in lung tissue compared to the nafamostat-treated group. Notably, no viral replication was detected in brain tissue when mice were treated with 60 mg/kg and 100 mg/kg of MDB-601a-NM. The glycyrrhizic acid-treated group showed no survival or inhibition of viral replication following SARS-CoV-2 infection. To assess histopathological differences, we analyzed H&E and IHC staining of collected tissue at 5 DPI. The nafamostat-treated group exhibited severe lesions, including pneumonia and a large number of antigens, similar to the MOCK-treated group. In contrast, the MDB-601a-NM-treated groups displayed improved histopathological changes and enhanced inhibitory efficacy, such as clear alveoli and respiratory veins and fewer antigens, depending on the MDB-601a-NM concentration.

However, our study has some limitations. It did not compare the antiviral efficacy of MDB-601a-NM with recently used COVID-19 antivirals, such as nirmatrelvir or molnupiravir. Moreover, no significant difference in viral replication was observed in lung and brain tissues. Therefore, combination therapy with MDB-601a-NM and other antivirals could potentially enhance its efficacy against SARS-CoV-2. Finally, the exploration of other antivirals with glycyrrhizic acid formulations could lead to the development of novel treatment options.

## 4. Materials and Methods

### 4.1. Preparation of Nafamostat-Loaded Micelles

To prepare the micelles, 20 mg of glycyrrhizin (DB13751; TOKYO CHEMICAL INDUSTRY, Tokyo, Japan) was dissolved in 1 mL of ethanol. Subsequently, 1 mL of oil was carefully added to the solution, ensuring proper layer separation. In parallel, 4 mg of nafamostat mesylate (EnzyChem Lifesciences, Seoul, Korea) was dissolved in 10 mL of distilled water. The ethanol layer containing glycyrrhizin was then combined with the nafamostat solution and sonicated using a probe-type sonifier. The resulting mixture was dialyzed using a dialysis membrane with a molecular weight cut-off of 6000–8000 Da, followed by lyophilization. The final product, nafamostat mesylate-loaded micelle nanoparticles (NPs), is referred to as NM-loaded micelle NPs.

### 4.2. Characterization of Nafamostat-Loaded Micelles

The morphology of both glycyrrhizin micelle NPs and NM-loaded micelle NPs was examined using a transmission electron microscope (TEM, Talos L120C, FEI, Prague, Czech) at the National Instrumentation Center for Environmental Management (NICEM). Each type of NP was separately dispersed in distilled water (0.1 mg mL^−1^) to measure their particle size distributions using an SZ-100V2 instrument (HORIBA Instruments Inc., Kyoto, Japan). The particle size of MDB-601-NM was determined independently through six measurements.

### 4.3. Cell Culture and Virus

Vero E6 cells (CRL1586) were obtained from the American Type Culture Collection (ATCC, Rockville, MD, USA). Cells were cultured in Dulbecco’s modified Eagle’s medium (DMEM, GIBCO, Gaithersburg, MA, USA) supplemented with 10% fetal bovine serum (FBS) and 1% antibiotic-antimycotic at 37 °C in a 5% CO_2_ incubator. SARS-CoV-2 was propagated in DMEM supplemented with 2% FBS and 1% antibiotic-antimycotic, and cultured for 96 h at 37 °C in a 5% CO_2_ incubator.

### 4.4. Pharmacokinetics and Single-Dose Toxicity of MDB-601a-NM

To analyze the pharmacokinetics of MDB-601a-NM, 8-week-old Sprague-Dawley rats were divided into individual groups, each consisting of 6 rats. A dose of 1 mg/kg nafamostat was intravenously (IV) administered to each rat, and doses of 20 mg/kg nafamostat and 50, 80, and 100 mg/kg of MDB-601a-NM were subcutaneously (SC) administered to each rat. Blood samples were collected at multiple time points (0.033, 0.088, 0.17, 0.25, 0.5, 1, 2, 4, 6, 8, 12, 24, 36, 48, 60, and 72 h post-administration). Serum was obtained from the blood samples and used for pharmacokinetics analysis (L2 Sciences, Ansan, South Korea). For single-dose toxicity analysis, six-week-old male and female Sprague-Dawley rats were divided into individual groups, each consisting of five rats. Each rat received a single subcutaneous dose of 800, 2000, or 5000 mg/kg of MDB-601a-NM or phosphate-buffered saline (PBS). Injection sites were monitored for swelling up to 14 days post-inoculation (DPI), and weights were recorded at 0, 3, 7, and 14 DPI. Clinical scoring on a scale of 1–5 assessed the condition of the administered rats. All single-dose toxicity tests were conducted at DT&Cro (Yongin, South Korea). The pharmacokinetics experiment was conducted in two stages: an initial pilot experiment followed by a subsequent main experiment. Each experiment was conducted independently, and the results from the second experiment were included.

### 4.5. In Vivo Efficacy Evaluation in K18-ACE2 Transgenic Mice

Eight-week-old female K18-hACE2 transgenic mice (B6.Cg-Tg(K18-ACE2)2Prlmn/J) were obtained from Jackson Lab (Bar Harbor, ME, USA). The Institutional Animal Care and Use Committee of Chungbuk National University (CBNUA-1646-21-01) approved all procedures. Following a seven-day acclimatization period, the mice were randomly assigned to one of seven experimental groups. The uninfected treatment control group (Vehicle) consisted of 12 mice, while each infected group contained 12 mice. The mice were anesthetized with isoflurane and intranasally inoculated with 50 μL of a viral solution containing 5 MLD50 of S clade SARS-CoV-2 (BetaCoV/Korea/KCDC03/2020) virus, isolated by the National Culture Collection for Pathogens of the Korea Center for Disease Control and Prevention. In the treatment groups, MDB-601a-NM at doses of 20 mg/kg (obtain 57 mg/kg of GA), 60 mg/kg (obtain 171 mg/kg of GA), or 100 mg/kg (obtain 285 mg/kg of GA), and glycyrrhizic acid at 285 mg/kg were subcutaneously administered 6 h post-infection. Subsequently, MDB-601a-NM was administered once daily every 24 h until 4 days post-inoculation (DPI), totaling 5 administrations. The nafamostat control group received a 100 mg/kg subcutaneous dose 6 h post-infection and was administered every 24 h until 4 DPI (5 total administrations). The mock-treated mice received saline, similar to the MDB-601a-NM and nafamostat groups. In the vehicle group, MDB-601a-NM was administered subcutaneously at 100 mg/kg without SARS-CoV-2 infection. On the fifth DPI, six mice per group were euthanized, and lung and brain tissues were collected. The tissues were stored at −80 °C for viral titer assays and inactivated with a 10% formalin solution (Sigma Aldrich, St. Louis, MI, USA) for histopathology and immunohistochemistry analyses. Weight changes and survival rates were recorded daily until 14 DPI. The efficacy evaluation was independently conducted in two individual experiments.

### 4.6. Viral Titration

To determine the viral titer, frozen tissues collected from infected mice at 5 DPI were homogenized in 500 μL of serum-free DMEM using Tissue Lyser II (Qiagen, Hilden, Germany). The resulting supernatants were serially diluted 10-fold in serum-free DMEM. Vero E6 cells were seeded at a density of 1 × 10^6^ cells per 96-well culture plate. After 24 h of incubation, the cells were inoculated with the serially diluted samples for 1 h. The inoculum was then replaced with fresh DMEM containing 2% FBS, and the cells were incubated for an additional 96 h. Following incubation, cells were stained with a 10% crystal violet solution, and the 50% tissue culture infective dose (TCID50) was calculated using the Reed–Muench method (1938).

### 4.7. Histology and Immunohistochemistry

Lung and brain tissues collected from mice were fixed with a 10% formalin solution (Sigma Aldrich), embedded in paraffin, and sectioned at a thickness of 3 μm. Hematoxylin and eosin (H&E) staining was performed on the tissue sections to evaluate histopathological findings. For immunohistochemical staining (IHC), SARS-CoV-2 (2019-nCoV) Nucleocapsid Antibody (40143-R019, SinoBiological, Beijing, China) was used as the primary antibody. The IHC staining procedure followed the protocol for the Ventana Discovery ULTRA stainer (Roche, Basel, Switzerland). All slides were visualized using a Pannoramic 250 Flash III slide scanner (3DHISTECH Ltd., Budapest, Hungary) and assessed with CaseViewer 2.1 software (3DHISTECH Ltd., Budapest, Hungary).

### 4.8. Statistical Analysis

Results were presented as mean ± standard deviation. One-way ANOVA was employed to determine statistical significance among groups, followed by Tukey’s multiple comparison tests. Survival curves were analyzed using the Kaplan–Meier method with the logrank (Mantel–Cox) test. GraphPad Prism software version 9 (San Diego, CA, USA) was utilized for statistical analysis. A *p*-value < 0.05 was considered statistically significant.

## Figures and Tables

**Figure 1 ijms-24-09579-f001:**
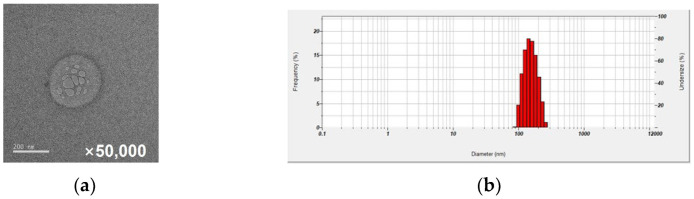
(**a**) To investigate the particle formation of glycylserylamine and the encapsulation of nafamostat, transmission electron microscopy (Talos L120C, FEI, Prague, Czech) was employed to observe MDB-601a-NM samples at a magnification of 50,000 times. (**b**) The particle size analysis of the synthesized MDB-601a-NM nanoparticles using a particle size analyzer (SZ-100V2, HORIBA instruments Inc., Kyoto, Japan). The x-axis at the bottom of represents the particle size, and the left y-axis represents the frequency distribution of particle sizes.

**Figure 2 ijms-24-09579-f002:**
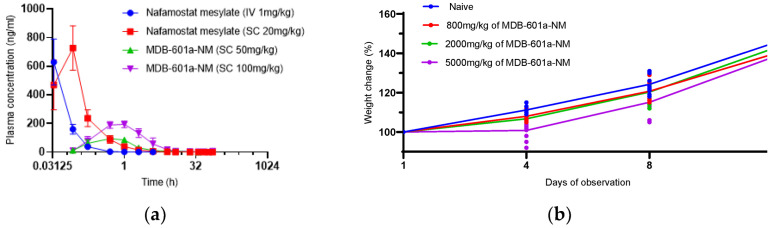
Pharmacokinetics and single dose injection of MDB-601a-NM, (**a**) Plasma pharmacokinetic profiles of MDB-601a-NM or Nafamostat mesylate in the mice at 0, 0.033, 0.088, 0.17, 0.25, 0.5, 1, 2, 4, 6, 8, 12, 24, 36, 48, 60, and 72 h. (**b**) Weight changes of 6-week-old male and female Sprague-Dawley rats after 800, 2000, or 5000 mg/kg of MDB-601a-NM single dose injection.

**Figure 3 ijms-24-09579-f003:**
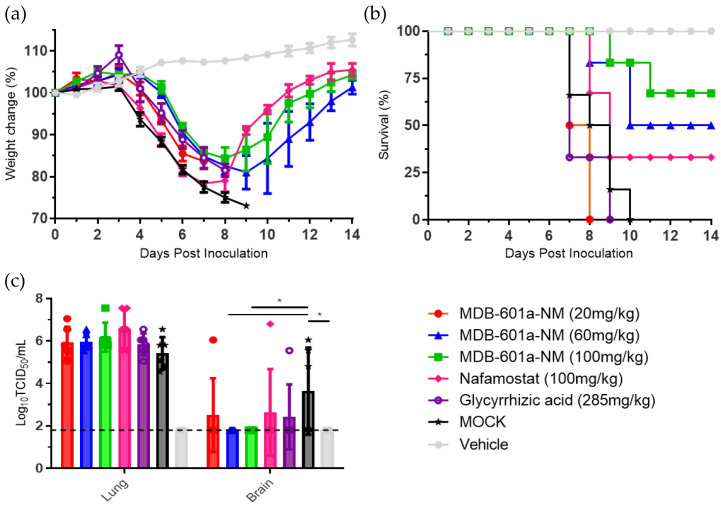
Evaluation of the therapeutic efficacy of MDB-601a-NM, nafamostat, and glycyrrhizic acid in SARS-CoV-2 infected K18-hACE2 transgenic mice. (**a**) Weight change was monitored daily for 14 DPI. (**b**) Survival rates. (**c**) Evaluation of the viral replications in lung and brain tissues of SARS-CoV-2 infected mice. One-way ANOVA with Tukey’s multiple comparison test; * *p* < 0.05.

**Figure 4 ijms-24-09579-f004:**
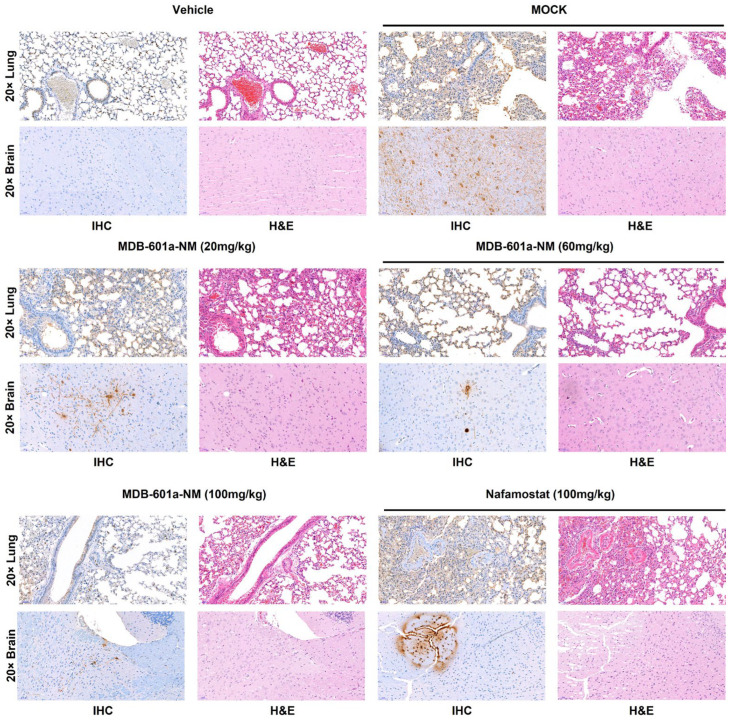
Histopathology and immunohistochemistry analysis of SARS-CoV-2 infected treated with MDB-601a-NM, nafamostat, and glycyrrhizic acid. Pathological changes and virus distribution were measured at 5 DPI in the lung and brain tissue by H&E staining and IHC staining with an anti-NP antibody. Images are shown at 20× power resolution.

**Table 1 ijms-24-09579-t001:** Observations of MDB-601a-NM injection sites.

Gender	Dose (mg/kg)	No. of Animals	Clinical Sign	Days of Observation													
				2	3	4	5	6	7	8	9	10	11	12	13	14	15
Male	Naïve	5	NOA ^a^	5	5	5	5	5	5	5	5	5	5	5	5	5	5
	800	5	NOA	5	5	5	4						1	2	2	4	4
			Swelling ^b^				1	5	5	5	5	5	4	3	3	1	1
	2000	5	NOA	5	5	2											
			Swelling			3	5	5	5	5	5	5	5	5	5	5	5
	5000	5	NOA	5	5												
			Swelling			5	5	5	5	5	5	5	5	5	5	5	5
Female	Naïve	5	NOA	5	5	5	5	5	5	5	5	5	5	5	5	5	5
	800	5	NOA	5	5	5	3							1	1	2	2
			Swelling				2	5	5	5	5	5	5	4	4	3	3
	2000	5	NOA	5	5	3											
			Swelling			2	5	5	5	5	5	5	5	5	5	5	5
	5000	5	NOA	5	5		1										
			Swelling			5	4	5	5	5	5	5	5	5	5	5	5

^a^ NOA: No observable abnormality. ^b^ Observations of swelling in the injection site.

## Data Availability

The data that support the findings of this study are available from the corresponding author upon reasonable request.

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
