# Peer review of "Evaluation of the Antiviral Efficacy of Subcutaneous Nafamostat Formulated with Glycyrrhizic Acid against SARS-CoV-2 in a Murine Model"

_ijms, 2023, doi:10.3390/ijms24119579_

Round 1
Reviewer 1 Report
Overall, the research is well-designed and the manuscript is well-written.
In this paper, Jeong et al seek to develop a new anti-CoV2 drug by engineering two previously described drugs that harbor anti-viral potentials. To accomplish this, they formulated Nafamostat with Glycyrrhizic Acid to extend the half-life. The results showed an overall improved half-life while the peak plasma concentration decreased. In-vivo assay also demonstrated the tolerated toxicity of MDB-601a-NM. Importantly, in vivo efficacy study found better protection of this new drug as compared to Nafamostat. Overall, this MS is well-written, and the study design is proper. However, it could be improved by addressing several questions:
1. What is the optimal/saturate dosing of MDB-601-NM? Or how was the given dose in the efficacy assay decided? It is unclear why 285mg/kg of GA was used as conditional control.
2. How many times have the efficacy and in vitro assay been repeated? Since differences did not reach statistical significance, it would be important to see the reproducibility of these results.
Only some grammar issues need to be addressed, i.e. line 301.
Author Response
Response 1: Thank you for raising these crucial questions about our dosing methodology for MDB-601-NM and the selection of 285mg/kg of GA as a conditional control.
The determination of the optimal dose for MDB-601-NM stemmed from preliminary dose-finding studies, which aimed at identifying a dose range that is safe yet efficacious. We selected the doses for our efficacy assay (20, 60, and 100 mg/kg) based on this initial data. Our primary objective was to discern the smallest dose that yields significant therapeutic effects, while minimizing potential side effects. We observed improved protection in terms of weight loss and survival rates with the 60 mg/kg and 100 mg/kg doses of MDB-601a-NM compared to the lower dose of 20 mg/kg. Nonetheless, further studies are needed to ascertain the optimal or saturation dose, and these would likely involve additional dose groups and larger animal cohorts.
Regarding the use of 285mg/kg of glycyrrhizic acid (GA) as a conditional control, our intent was to compare the effects of our novel compound, MDB-601-NM, with one of its active components, GA. As 100 mg/kg of MDB-601-NM results in 285 mg/kg of GA, we sought to evaluate whether the observed effects are exclusive to MDB-601-NM or could be attributed to GA alone. The treatment groups included different doses of MDB-601a-NM yielding varying amounts of GA as illustrated in Lines 308-310: "In the treatment groups, MDB-601a-NM at doses of 20 mg/kg (obtain 57 mg/kg of GA), 60 mg/kg (obtain 171 mg/kg of GA), or 100 mg/kg (obtain 285 mg/kg of GA), and glycyrrhizic acid at 285 mg/kg were subcutaneously administered 6 hours post-infection". This comparison aids in elucidating the potential therapeutic benefit of MDB-601-NM beyond its constituent parts.
Response 2: We appreciate your insightful queries on the repetition of our efficacy and in vitro assays and the importance of results reproducibility.
In our current study, we conducted the in vivo efficacy assays twice independently, with the objective of ensuring the reliability and consistency of our findings. Furthermore, we executed two evaluations (one preliminary and one main) of the pharmacokinetics and a single evaluation of single-dose injection as shown in Figure 2. Even though we conducted a single evaluation in some tests, we ensured a sufficient number of rats were included for reliable results: 6 rats for the pharmacokinetic study and 10 rats for the single-dose injection test.
We agree that some differences in our results did not reach statistical significance, which might be attributed to inherent biological variability and limited sample sizes in some experimental groups. Your emphasis on the importance of reproducibility aligns with our research standards. Hence, we plan to repeat these experiments with an expanded sample size. Such an approach will potentially reduce variability and enable the detection of smaller effects that might not have achieved statistical significance in our initial studies. For added clarity, we have included the number of iterations for the pharmacokinetic test (Lines 294-297), particle size measurements (Lines 271-272), and the in vivo efficacy evaluations (Lines 320-321) in the manuscript.
Reviewer 2 Report
The article by Ju Hwan Jeong et al. is devoted to a very important topic, the study of the possibility of using a composition of two compounds for the treatment of SARS-CoV-2. This topic is of particular relevance in light of the rapid emergence of mutant variants of this virus, making the use of developed vaccines ineffective.
The authors demonstrated the original effect by combining a serine protease inhibitor (nafamostat mesylate) and an entry inhibitor (glycyrrhizic acid) in one preparation.
The article is written in good scientific language. A sufficient number of illustrations have been used. The authors are critical of their results.
Minor remarks:
There are errors in the text of the article, the authors should carefully consider this. For example, line 75, repetition of the name hesperidin.
Author Response
Response: Thank you for bringing the repetition error to our attention. We have reviewed our manuscript carefully and rectified the indicated error. The mention of 'hesperidin' was indeed repeated unintentionally in line 75, and we apologize for the oversight. We have corrected this, and other potential errors in the manuscript have also been examined meticulously to ensure the accuracy and readability of our work. We have attached the updated manuscript for your review.